# Automatic multi-organ segmentation in dual energy CT using 3D fully convolutional network

**Shuqing Chen**
Pattern Recognition Lab
Department of Computer Science
Friedrich-Alexander-Universität
Erlangen-Nürnberg
`shuqing.chen@fau.de`

**Xia Zhong**
Pattern Recognition Lab
Department of Computer Science
Friedrich-Alexander-Universität
Erlangen-Nürnberg

**Shiyang Hu**
Pattern Recognition Lab
Department of Computer Science
Friedrich-Alexander-Universität
Erlangen-Nürnberg
Erlangen Graduate School
in Advanced Optical Technologies (SAOT)

**Sabrina Dorn**
German Cancer Research Center (DKFZ)
Ruprecht-Karls-University Heidelberg

**Marc Kachelrieß**
German Cancer Research Center (DKFZ)
Ruprecht-Karls-University Heidelberg

**Michael Lell**
University Hospital Nürnberg
Paracelsus Medical University

**Andreas Maier**
Pattern Recognition Lab
Department of Computer Science
Friedrich-Alexander-Universität
Erlangen-Nürnberg
Erlangen Graduate School in Advanced Optical Technologies (SAOT)

## Abstract

Automatic multi-organ segmentation of the dual energy computed tomography (DECT) data is beneficial for biomedical research and clinical applications. Numerous recent researches in medical image processing show the feasibility to use 3-D fully convolutional networks (FCN) for voxel-wise dense predictions of medical images. In the scope of this work, three 3D-FCN-based algorithmic approaches for the automatic multi-organ segmentation in DECT are developed. Both of the theoretical benefit and the practical performance of these novel deep-learning-based approaches are assessed. The approaches were evaluated using 26 torso DECT data acquired with a clinical dual-source CT system. Six thoracic and abdominal organs (left and right lungs, liver, spleen, and left and right kidneys) were evaluated using a cross-validation strategy. In all the tests, we achieved the best average Dice coefficients of 98% for the right lung, 97% for the left lung, 93% for the liver, 91% for the spleen, 94% for the right kidney, 92% for the left kidney, respectively. Successful tests on special clinical cases reveal the high adaptability of our methods in the practical application. The results show that our methods are feasible and promising.

1st Conference on Medical Imaging with Deep Learning (MIDL 2018), Amsterdam, The Netherlands.

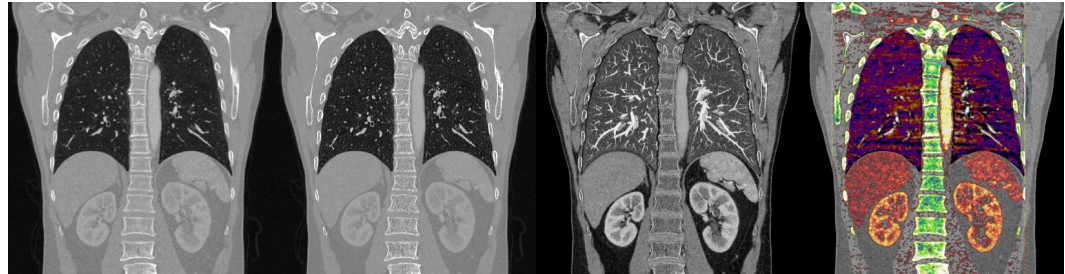

(a) conventional reconstruction    (b) CS reconstruction      (c) CS display      (d) CS dual energy

Figure 1: Examplary application of multi-organ segmentation: context-sensitive (CS) DECT imaging. (a): Coronal view of a conventional CT image for comparison purposes. (b): Context-sensitive reconstruction with an organ-adapted spatial resolution and noise level. (c): The context-sensitive image is shown in a organ-adapted gray level window (lung in lung window, soft tissue in body window, bone in bone window, etc.) and with a sliding thin slab (STS) mean intensity projection in the lung and STS maximum intensity projection in the lung simultaneously. (d): Color overlay of multiple dual energy evaluation (liver iodine quantification, body iodine quantification, lung perfused blood volume, ect.) that are applied in an organ-specific manner.

## 1 Introduction

In computed tomography (CT), the Hounsfield unit (HU) scale value depends on the inherent tissue properties, the X-ray spectrum for scanning, and the administered contrast media [1]. Materials, having different elemental compositions, can be represented by identical HU values in a SECT image [2]. Therefore, single energy CT (SECT) has the characteristics of providing limited material-specific information and beam hardening as well as tissue characterization [1]. These characteristics influence the organ segmentation. DECT has been investigated to overcome the shortcomings of SECT. In DECT, two energy-specific image data sets are acquired using two different X-ray energy spectra simultaneously. The two energy-specific image datasets facilitate the differentiation of tissue materials and therefore the multi-organ segmentation can be improved. The exploited prior anatomical information gained through the multi-organ segmentation, can be used to improve biomedical research and clinical applications, such as material decomposition [3] and context-sensitive DECT imaging [4–6] (example see Fig. 1). The novel technique offers the possibility to present evermore complex information to the radiologists simultaneously and provides the potential to improve the clinical routine in CT diagnosis.

Automatic multi-organ segmentation on DECT images is a challenging task due to the inter-subject variance of human organs, the soft anatomy deformation, as well as different HU values for the same organ by different spectra. Numerous recent researches show the successful experience of deep learning in single or multiple organ segmentation using volumetric SECT images [7, 8]. Our prior study shows that deep learning techniques are also feasible for the DECT image segmentation [9]. In this prior study, the image information of two energies is fused in the preprocessing based on a linear weighted sum function. The experiments showed that the segmentation with optimal fusion weight achieved better accuracy than using one single energy image. However, the manual weight selection influences the segmentation accuracy and is organ-dependent. Thus, the spectral information needs to be exploited more efficiently and automatically by e.g. the implementation of the image fusion into the network.

In this work, we propose three 3D deep learning architectures for automatic multi-organ segmentation on DECT images. The proposed methods are all based on an end-to-end 3D FCN [10]. The energy-dependent anatomical information of two energies is fused in the network while neither mixture weight nor organ-specific or energy-specific apriori knowledge is required. To solve the high class imbalance problem of multi-organ segmentation, a loss function based on weighted Dice coefficient is designed. The cross-validation results show that these proposed methods are promising to solve multi-organ segmentation problems for DECT and outperform the prior work.

## 2 Materials and methods

### 2.1 Network architectures for DECT organ prediction

In clinical diagnose using DECT, a mixed image display is employed as described by Krauss et al. [11]. The calculation of the mixed image is a linear weighted addition of the two images acquired by the two spectra:

$$I_{\text{mix}} = \alpha \cdot I_{\text{low}} + (1 - \alpha) \cdot I_{\text{high}} \tag{1}$$

where $\alpha$ is the weight of the dual energy composition. $I_{\text{mix}}$ denotes the mixed image. $I_{\text{low}}$ and $I_{\text{high}}$ are the images at low and high tube voltage, respectively.

Our prior study [9] shows that segmentation accuracy is dependent on the $\alpha$ and the optimal $\alpha$ value is organ-dependent. The $\alpha$ value must be selected manually in the prior study and the mixture is a preprocessing step outside the FCN. Instead of considering the mixture of two energy channels as preprocessing, we could embed this step into the network. Using this embedding, we could avoid the manual selection of $\alpha$ and exploit the optimal mixture of the spectral information more efficiently and automatically.

One possible approach is based on the linear combination shown in Eq. 1 directly, and is named as **linear combination based network (LC-net)**. The LC-net uses the alpha blending to merge images of two energy levels. This can be implemented as merge layer with a single trainable parameter. To facilitate the implementation in the FCN, we derive the Eq. 1 to Eq. 2. Fig. 2a illustrates the architecture of this approach. Using the dual energy data as two inputs of the FCN, the input images are merged with mathematical layers and the mixed image is sent into the FCN. At the final stage of the expansive path, the extracted features from the original inputs are also concatenated into the upsampled image in order to add more details.

$$I_{\text{mix}} = I_{\text{high}} + \alpha \cdot (I_{\text{low}} - I_{\text{high}}) \tag{2}$$

However, the LC-net may be too general for the image fusion according to the principle of CT image composition. Using additive noise model, a CT image $I$ can be expressed as $I = I_{\text{inf}} + P(E) + \mathcal{N}(0)$ [12] where $I_{\text{inf}}$ denotes the noise-free image, $P(*)$ denotes the Poisson noise, $E$ denotes the energy spectrum, and $\mathcal{N}$ denotes the white Gaussian noise. Theoretically, the Poisson noise of CT depends on the energy of the photon and the number of the photon (i.e. detector entrance dose and spectrum). Assuming the detector has the same white noise, i.e. the identical normal distribution. The mixture of DECT image Eq. 1 can be expressed as Eq. 3. As we could see, the simple linear weighted mixture merges not only the image signal but also the noise. A solution for image fusion with noise elimination is expected consequently.

$$I_{\text{mix}} = \alpha \cdot I_{\text{inf}_{\text{low}}} + (1 - \alpha) \cdot I_{\text{inf}_{\text{high}}} + \alpha \cdot P(E_{\text{low}}) + (1 - \alpha) \cdot P(E_{\text{high}}) + \mathcal{N}(0) \tag{3}$$

Another intuitive way of merging information in the FCN is to consider the two images of different energies as different channels of an input image. This solution is named as **multi-channel based network (MC-net)**, and is shown in Fig. 2b. The two-channel FCN uses only one image input and employs the dual energy data as two channels of the FCN input. This network architecture processes two energy spectra using different kernels in the first convolutional layer and subsequently added the filtered images. This merge in a single channel (before activation) can be established as Eq. 4, where $*$ denotes the convolution, $W_a$ and $W_b$ denotes different kernel. The mixture of the information happens in the low-level feature maps and consequently, the noise still remains in the mixture.

$$I_{\text{mix}} = W_a * I_{\text{inf}_{\text{low}}} + W_b * I_{\text{inf}_{\text{high}}} + W_a * P(E_{\text{low}}) + W_b * P(E_{\text{high}}) + W_a * \mathcal{N}(0) + W_b * \mathcal{N}(0) \tag{4}$$

The further possibility is to branch the FCN in the feature extraction path. This solution is named as **Multi-image based network (MI-net)** and is illustrated in Fig. 2c. The dual energy data is employed as two inputs of the FCN. The two input images are handled separately by the two branches of the FCN to extract the high-level features. The merge of information is implemented using the features at the highest level only. In the concatenation of each depth, the features of the two outputs of the contracting path are employed. In this solution, the energy noise of each image can be handled separately. It neither mixes the noise in the low level nor propagates the mixture of the noise in the network, therefore the trained network should be more robust against the noise.

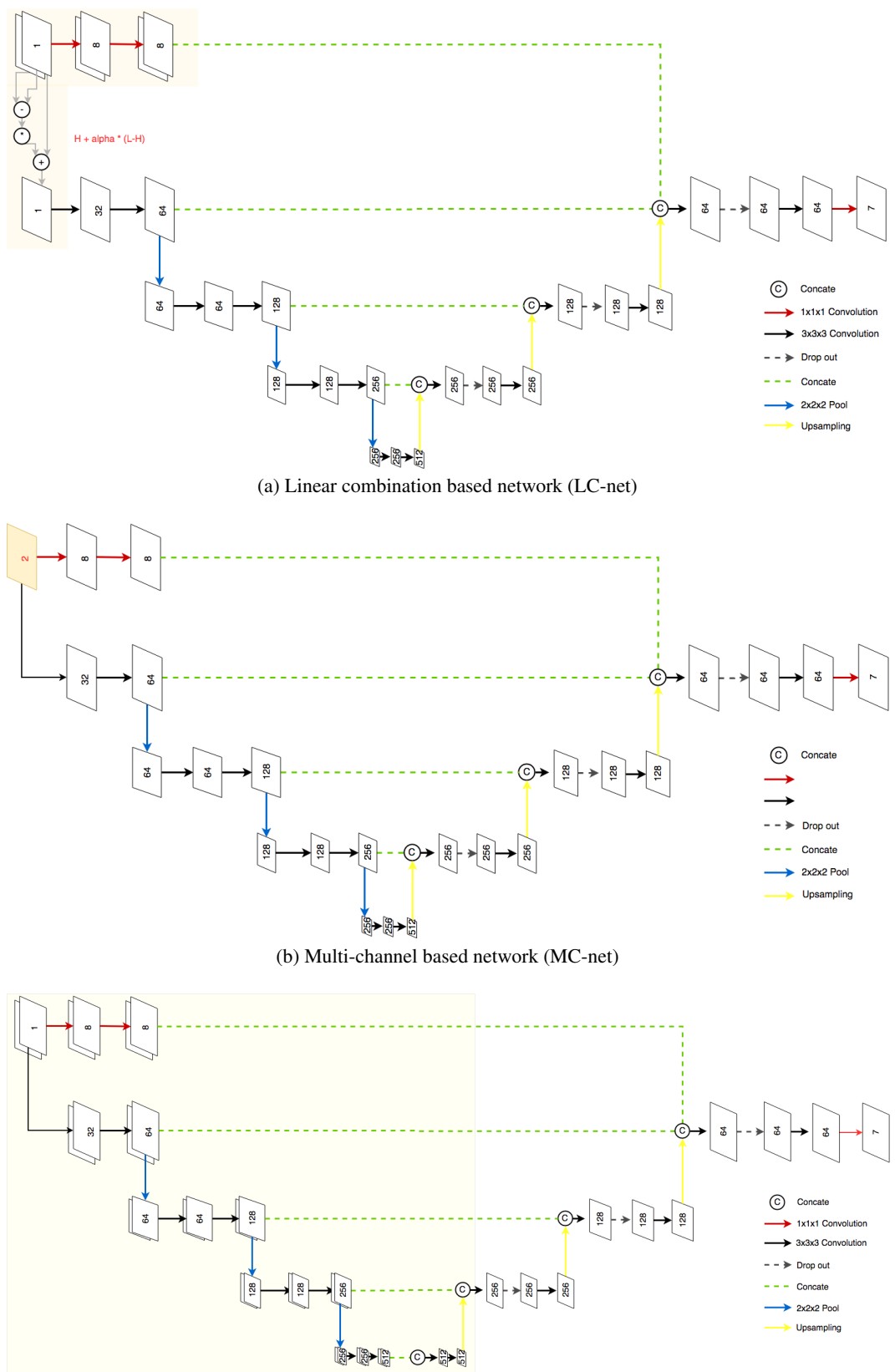

(a) Linear combination based network (LC-net)

(b) Multi-channel based network (MC-net)

(c) Multi-image based network (MI-net)

Figure 2: Network architectures of DECT multi-organ segmentation. Differences are highlighted

## 2.2 Loss function

The three models are all trained with softmax as activation and weighted multi-class Dice as loss. For each class $c$, given the estimated probability $p_c(\mathbf{x})$ and the ground truth probability $g_c(\mathbf{x})$ at voxel $\mathbf{x}$, the loss function is defined as:

$$L = -w_c(\mathbf{x}) \cdot \frac{2\sum_{\mathbf{x}} p_c(\mathbf{x})g_c(\mathbf{x})}{\sum_{\mathbf{x}} p_c^2(\mathbf{x}) + \sum_{\mathbf{x}} g_c^2(\mathbf{x})} \tag{5}$$

Because the voxel amounts of the kidneys are much less than the ones of the other organ, this multi-organ segmentation is high class imbalance, the multi-class Dice in the loss function should be therefore balanced. A median frequency balancing is used in the class weights $w_c(\mathbf{x})$ of the Dice loss to overcome the class imbalance problem. Furthermore, to correct the boundary errors in the segmentation, a boundary compensation introduced in [13] is also implemented in the class weights. The class weights $w_c(\mathbf{x})$ are thus composed of two terms: the median frequency balancing and the boundary compensation, and given as:

$$w_c(\mathbf{x}) = \frac{median(\mathbf{f})}{f_c} + \lambda \cdot I(|\bigtriangledown G_c(\mathbf{x})| > 0) \tag{6}$$

where scalar $f_c$ denotes the frequency of class $c$ in the training data, i.e. the class probability. The vector $\mathbf{f}$ is the vector of all frequencies. $I$ is the indicator function. $G_c$ is the ground truth of class $c$. The operator $\bigtriangledown$ denotes the 2-D gradient operator. The median frequency balancing term is used to highlight classes with low probability for the compensation of the class imbalance. The boundary compensation term increases the weights on the anatomical boundaries for contour correction. $\lambda$ balances the two terms.

## 2.3 Model learning

These proposed networks were evaluated with 26 clinical torso DECT images scanned by the department of radiology, university hospital Erlangen, using a Siemens SOMATOM Force CT system. The images were taken from male and female adult patients with different clinically oriented indication justified by the radiologist. All of the images were acquired using the same scan protocol: X-ray tube voltage settings of 70 kV (560 mAs) and Sn 150 kV, ultravist 370 as the contrast agent, and kernel Q32 for the image reconstruction. The volumes contain 992-1248 slices with slice size 512x512 pixels, and voxel spacing from [0.6895-0.957, 0.6895-0.957,0.6] mm are used. Six abdominal and thoracic organs were tested, including liver, spleen, right and left kidneys, and right and left lungs. Ground truths were generated manually by experts. To improve the accuracy of the ground truths they were reviewed and refined by different experts.

The segmentation accuracy was evaluated with 9-fold cross-validation. For the selection of training, validation and test datasets, we showed that the bias of random data selection decreases the segmentation accuracy [14]. Therefore, we integrated the manifold learning-based technique proposed in [14] to split the data into training, validation, and test datasets, in order to keep the distributions of the datasets similar. The images were projected into 2-D space using locally linear embedding (LLE) [15] and clustered into three classes using k-means. Training data, validation data, and test data were selected randomly within the three classes with the ratio about 7:1:1. That means for one evaluation scenario, 20 images were used for training, 3 images for validation and 3 for test.

The original images were downsampled by the factor of two. Afterward, the images were split into patches with size [32, 32, 32] and overlap 16. To increase the data quantity, the patches are augmented with translation, rotation, scaling and elastic deformation [16]. Finally, we generated 27465 patches for training and 3715 patches for validation.

We used Adam as the optimizer. The learning rate is initially set to 0.00001 and reduced by factor 0.1 after every 10 epochs till convergence. NVIDIA TITAN Xp with 12 GB memory was used for learning, which constrained the batch size of 6 for training and 3 for validation.

Table 1: Dice score comparison of different approaches.

| | Right Lung | Left Lung | Right Kidney | Left Kidney | Liver | Spleen |
|---|---|---|---|---|---|---|
| LC-net | $0.97 \pm 0.01$ | $\mathbf{0.97 \pm 0.01}$ | $\mathbf{0.93 \pm 0.02}$ | $\mathbf{0.91 \pm 0.05}$ | $\mathbf{0.92 \pm 0.02}$ | $0.88 \pm 0.06$ |
| MC-net | $0.97 \pm 0.01$ | $\mathbf{0.97 \pm 0.01}$ | $\mathbf{0.93 \pm 0.02}$ | $0.84 \pm 0.15$ | $0.91 \pm 0.02$ | $0.87 \pm 0.07$ |
| MI-net | $\mathbf{0.98 \pm 0.01}$ | $\mathbf{0.97 \pm 0.01}$ | $0.93 \pm 0.03$ | $\mathbf{0.91 \pm 0.05}$ | $\mathbf{0.92 \pm 0.02}$ | $\mathbf{0.91 \pm 0.04}$ |
| Best with manual $\alpha$ in [9] | $0.96 \pm 0.02$ | $0.96 \pm 0.01$ | $0.91 \pm 0.03$ | $0.89 \pm 0.05$ | $0.93 \pm 0.01$ | $0.92 \pm 0.03$ |
| Worst with manual $\alpha$ in [9] | $0.90 \pm 0.11$ | $0.89 \pm 0.10$ | $0.75 \pm 0.08$ | $0.76 \pm 0.23$ | $0.87 \pm 0.10$ | $0.82 \pm 0.11$ |

Table 2: Hausdorff distance (in voxel) comparison of different approaches.

| | Right Lung | Left Lung | Right Kidney | Left Kidney | Liver | Spleen |
|---|---|---|---|---|---|---|
| LC-net | $4.68 \pm 1.71$ | $4.54 \pm 1.78$ | $2.27 \pm 1.75$ | $3.59 \pm 1.47$ | $8.94 \pm 5.17$ | $4.76 \pm 1.86$ |
| MC-net | $4.90 \pm 1.93$ | $\mathbf{4.27 \pm 0.97}$ | $3.04 \pm 1.92$ | $6.90 \pm 7.14$ | $10.25 \pm 3.49$ | $4.86 \pm 2.24$ |
| MI-net | $\mathbf{3.88 \pm 1.20}$ | $4.43 \pm 2.58$ | $\mathbf{1.97 \pm 1.08}$ | $\mathbf{2.78 \pm 1.16}$ | $\mathbf{8.87 \pm 3.69}$ | $\mathbf{4.46 \pm 2.71}$ |
| Best with manual $\alpha$ in [9] | $4.20 \pm 1.32$ | $4.36 \pm 1.41$ | $3.18 \pm 3.93$ | $4.53 \pm 3.33$ | $8.93 \pm 3.88$ | $7.09 \pm 2.31$ |
| Worst with manual $\alpha$ in [9] | $6.90 \pm 2.11$ | $7.89 \pm 3.10$ | $6.39 \pm 8.69$ | $5.19 \pm 2.43$ | $9.32 \pm 9.88$ | $9.77 \pm 2.89$ |

## 3 Experiments and results

### 3.1 Comparison of architectures

**Accuracy estimation** The similarity between the segmentation result and the ground truth was measured using the Dice metric and the Hausdorff distance. We first compared the different architectures with $\lambda = 1$. Table 1 summarizes the average Dice scores on test data. The MI-net shows the best performance. Compared to the results with the optimal preprocessed fusion weight in our prior work [9], we achieved an average increase of $2\%$ for right lung, $1\%$ for left lung, and $2\%$ for both kidneys. However, the Dice scores for liver and spleen are $1\%$ lower. Table 2 lists the average of the Hausdorff distance for further comparison. The MI-net has lowest average Hausdorff distance and lowest standard deviation for most target organs, the improvement is significant. The comparison of the Hausdorff distance reveals that the MI-net overperforms our prior work.

**Performance comparison** Table 3 lists performance metrics of the networks with the same data selection and identical training configuration (see Sec. 2.3). Although the amount of the trainable parameters of the MI-net is almost 3 times higher as for the other two networks. Nonetheless, there is a tiny difference of the GPU occupancy rate and the epoch amount by the early stop. The time consumption is positively correlated with the amount of the trainable parameters but not linear proportional. Fig. 3 shows that the convergence efficiency of the three networks, by training and validation, related to epochs are all similar. Compared to the differences among the networks, the differences among organs are more significant.

### 3.2 Analyse of the balance weight $\lambda$

As shown in Eq. 6, we used the weight $\lambda$ to balance the compensation of the class imbalance and the correction of the anatomical boundaries. The effect of the $\lambda$ was also researched. We retrained MI-net with different $\lambda$. Table 4 lists the average Dice scores. The change of the Dice scores of both lungs is negatively proportional to the change of $\lambda$. However, the Dice scores of both kidneys increase when the $\lambda$ raises. The Dice scores of liver and spleen change up and down. The results show that the effect of $\lambda$ varies for organs. Small $\lambda$ provides higher results for lungs, while large $\lambda$ fits better for kidneys. The comparison also shows that, by changing the $\lambda$, the segmentation of the small organs change much less than the large organs. Small organs are more stable than large organs for $\lambda$ modification.

Table 3: Performance comparison of the three networks with the same training and validation data.

|  | Trainable parameters | GPU | Early stop | Total training time |
|---|---|---|---|---|
| LC-net | 16,343,800 | 11.505GB | 136 epochs | 12h 22min |
| MC-net | 16,330,759 | 11.427GB | 130 epochs | 11h 23min |
| MI-net | 46,925,815 | 11.429GB | 140 epochs | 13h 27min |

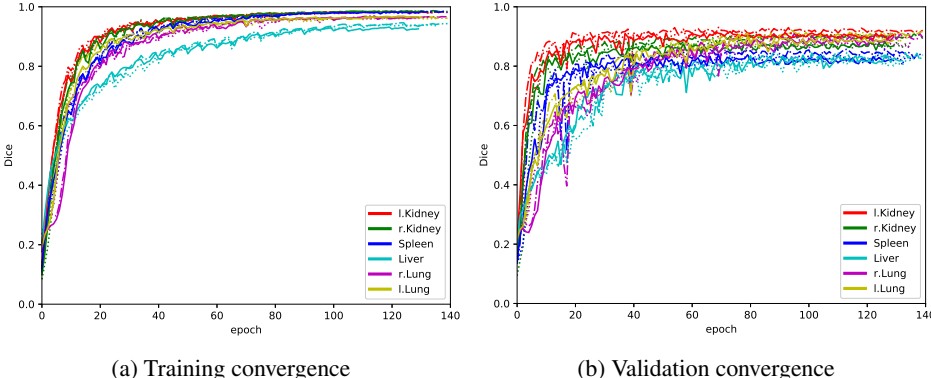

(a) Training convergence        (b) Validation convergence

Figure 3: Convergence comparison. Solid line denotes MI-net, dotted line denotes LC-net, and dash-dotted line denotes MC-net.

### 3.3 Robustness on special cases

To examine the robustness of these networks, we tested some special unseen clinical cases. The change of the acquisition protocol, such as different tube voltage and reconstruction kernel, can influence the images. Images with different scan protocol were tested. Also, images with obvious visual indication and images with missing organ or incomplete organ were experimented. Fig. 4a shows the successful prediction for an image acquired by 100kV and Sn 150kV and reconstructed with Qr40. Fig. 4b illustrates the accurate result of an image with obvious lung pathology. Fig. 4c presents the robustness of the proposed method applied to an image without the spleen. Fig. 4d shows that the proposed method also works great for truncated scan.

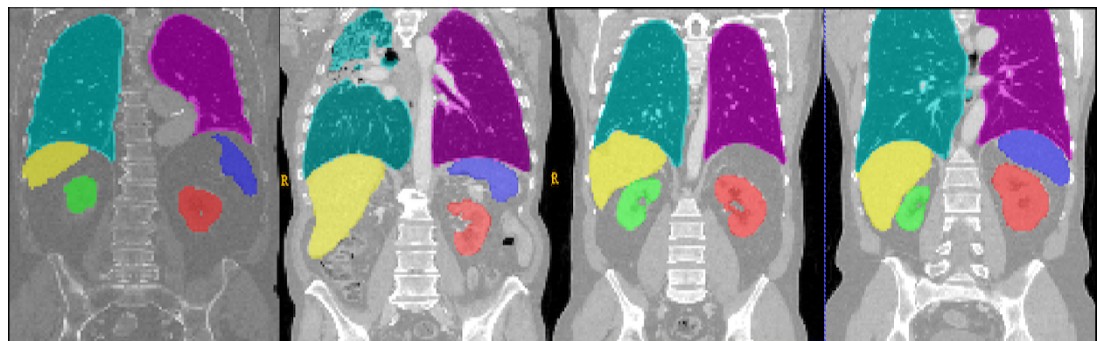

(a) Scan with different proto-col    (b) Scan with lung pathology    (c) Scan with missing spleen    (d) Scan with longitudinal truncation

Figure 4: Test examples of special unseen clinical cases with MI-net.

## 4 Discussion and conclusion

Our prior work in [9] shows that image fusion is an important part of the multi-organ segmentation on DECT images using deep learning. We proposed and compared three possible network architectures

Table 4: Result comparison of different $\lambda$ using MI-net.

| $\lambda$ | Right Lung | Left Lung | Right Kidney | Left Kidney | Liver | Spleen |
|---|---|---|---|---|---|---|
| 0 | $0.97 \pm 0.01$ | $\mathbf{0.97 \pm 0.01}$ | $0.90 \pm 0.08$ | $0.90 \pm 0.04$ | $0.90 \pm 0.03$ | $0.87 \pm 0.05$ |
| 1 | $\mathbf{0.98 \pm 0.01}$ | $\mathbf{0.97 \pm 0.01}$ | $0.93 \pm 0.03$ | $0.91 \pm 0.05$ | $0.92 \pm 0.02$ | $\mathbf{0.91 \pm 0.04}$ |
| 2 | $0.95 \pm 0.01$ | $0.96 \pm 0.02$ | $0.93 \pm 0.03$ | $0.91 \pm 0.05$ | $0.91 \pm 0.01$ | $0.88 \pm 0.05$ |
| 5 | $0.61 \pm 0.10$ | $0.53 \pm 0.06$ | $\mathbf{0.94 \pm 0.01}$ | $\mathbf{0.92 \pm 0.03}$ | $\mathbf{0.93 \pm 0.02}$ | $\mathbf{0.91 \pm 0.04}$ |

in this work, named LC-net, MC-net, and MI-net, respectively. LC-net uses alpha blending to merge the image information of two energies, however, the noise is also mixed with the alpha blending. Alpha blending is not able to handle the noise individually. MC-net merges images when images are filtered in the first subsequent convolution layer, i.e. the net uses a low-level mixture. Therefore, the mixture still contains noise and the noise is also propagated into the network. MI-net combines the important image information using high-level features. Therefore, the noise impact can be reduced at utmost, which increases the segmentation accuracy and robustness. The evaluation with Dice score and Hausdorff distance on unseen test data varified the theoretical analysis. The MI-net had the best performance on the Dice scores, also the improvement of the Hausdorff distance by using MI-net was significant. The compasion of training performance shows that the MI-net still provides efficient performance compared to the LC-net and the MC-net, although the trainable parameters are mutiplied. The comparison to our prior work shows that the three proposed networks are feasible and promising. Especially, the MI-net overperforms the prior approach. Moreover, the new methods are much easier to be fine-tuned than the prior approach.

The analyse of the balance weight $\lambda$ shows that the balance weight has more influence on large organs than small organs. The reason could be that the balance weight is added to the term of boundary correction. Compared to the large organs, the anatomical boundary of the small organs against organ size accounts for a greater proportion. Hence, for the small organs, the increase of the class weight due to the growth of the boundary weight is more significant than the large organs. With increasing $\lambda$, the loss function leans towards the smaller organs. This leads to high accuracy for small organs and low accuracy for the large organs. This analyse indicates that the loss function can be further improved in future. Dynamic weight may be introduced to explore the property of $\lambda$.

Furthermore, the success in various special unseen clinical cases shows that the proposed approaches are robust to special clinical cases, which reveals that the proposed methods have high adaptability in practical application.

### Acknowledgments

We gratefully acknowledge the German Research Foundation (DFG) through research grant No. KA 1678/20, LE 2763/2-1 and MA 4898/5-1 for the support of the work, and the NVIDIA Corporation for the donation of the NVIDIA TITAN Xp GPU.

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
