# OpenReview forum: "Automatic multi-organ segmentation in dual energy CT using 3D fully convolutional network"
_MIDL.amsterdam/2018/Conference — MIDL 2018 Poster_

### Review · AnonReviewer3 · 2018-05-09
**The authors presented three 3D FCNs for multi-organ segmentation in dual energy computed tomography (DECT). The three proposed methods are quite similar to each other. However, these methods have achieved better performance as compared to an existing method based on a closely related network.**

**Rating:** 2
**Confidence:** 2

**Review:**

Originality:
The proposed methods are mainly based on existing 3D FCNs. However, the authors made modifications based on their integration strategies. They proposed three different ways to integrate images from two different energy specific spectra.

Clarity:
Although most of the manuscript is easy to understand the network diagram lacks clarity. It is hard to map the equations in the Methods section to respective network diagrams.

Also, the description of Figure 1 is quite confusing. It would be better to split the description into subsections.

Quality:
One of the proposed methods (MI-net) outperformed the existing methods in terms of both Dice score and Hausdorff distance metric. However, the results of LC-net are better than the MC-net which is opposite to the justification presented in the Methods section.

Significance:
The proposed methods are evaluated on a very small and imbalance dataset of 26 voxels. However, they presented few visual results on challenging images to show the robustness of their methods.


**Special Issue:**

No

---

> ### Comment · ~Shuqing_Chen1 · 2018-05-16
> **Thank you for your review**
>
> Thank you very much for your review.
> About the data, we used data augmentation to increase the data quantity,  and we generated 27465 patches for training and 3715 patches for validation.
> We will consider and try to improve the diagrams to make them more clear and add more explanation about the results of LC-net and the MC-net in the later version.

---

### Review · AnonReviewer1 · 2018-05-09
**solid comparison, dual energy CT is interesting!**

**Rating:** 4
**Confidence:** 3

**Review:**

Summary of the paper:

This paper propose 3 approaches to solve the problem of dual energy CT multi-organ segmentation label fusion, called LC-Net, MC-Net and MI-Net. They demonstrate the effectiveness of the proposed systems using Dice score and Hausdorff distance in a small dataset of 20 scans.

Pros:

- The paper is very well written and structured, overall interesting paper
- Comprehensive comparison of experiments between the proposed 3 architectures and previous published work.
- The experimental settings, hyperparameters, and method details are very well outlined.
- The maths and the networks design are well explained.

Cons:

- The experimental results for the balance weight are inconclusive and a bit confusing, further experiments may give a better understanding of this parameter
- It would be advantageous to have a visual comparison of the ground truth and the MINet in Figure 4, rather than only the results.
- it would be a really complete paper if they show the results from experiments where only single energy is used for segmenting and compare whether dual energy CT add value to the fnal segmentation.

Constructive comments for the authors:

Overall good paper and suitable for the conference.

The size of the dataset is really small 20 scans to train, 3 validation and 3 test. Therefore,
it is a big claim to mention robustness and generalization with such a small dataset.

Somehow, Figure 1 caption and the way it is used in the text is not related.

**Special Issue:**

Yes

---

> ### Comment · ~Shuqing_Chen1 · 2018-05-14
> **Thank you for your review**
>
> Thank you very much for your review.
>
> About the comparison with SECT, yes it is compared. The comparison is described in the prior work, which is accepted by CT Meeting 2018 and the draft is preprinted by https://arxiv.org/abs/1710.05379. The prior work showed that the segmentation with optimal alpha achieved better accuracy than using one single energy image. I will write more about this prior work in the later version.
>
> Also thanks very much for other comments.  The parts mentioned will also be improved later. Thank you!

---

### Review · AnonReviewer2 · 2018-05-09
**Deep learning for Dual Energy CT**

**Rating:** 4
**Confidence:** 2

**Review:**

The paper presents and compares three possible approaches to dual energy CT analysis, one optimizing the mixing parameter of a linear combination simultaneously with a segmentation network, and two others performing early and late fusion of the information of the two input images, all using a similar U-net-like base network. Such DECT-specific are relevant and --- as far as I know --- novel.  Experiments are sufficient to demonstrate the strengths of the different approaches. The authors claim (statistically?) significant differences between approaches and results look good overall, also in difficult cases with e.g. a missing spleen or large abnormalities in the lungs.

This reviewer is not fully convinced of the importance of DECT compared to single energy CT for the application investigated in this paper (segmentation of lungs, spleen, liver, kidneys). But the paper does provide a nice comparison of the different approaches that should carry over to other DECT analysis problems.

One concern is that the dataset is relatively small, 26 images in total. The abstract mentions experiments are performed in cross-validation, which would be preferred, but Section 2.3 suggests train, validation, and test images were selected randomly with only 3 for testing. Please clarify. Also, are the claimed improvements statistically significant? Please specify performed statistical test(s) and p-values.

Another concern is what is the effect of the first skip connection in the MC-net, which is according to Figure 2 different from that in the LC- and MI-nets (LC and MI have 2 layers of 8 feature maps for both images, while MC has 2 layers of 8 feature maps for a single, already fused image). Perhaps the simplicity of this first step could already explain performance difference between MI and MC, especially since the otherwise simpler LC performs much better than MC?

Minor comments:
•	Performance improvement by the MI-net is most notable in the spleen. Do you have an explanation for that?
•	What is the number of filters and filter size in the different layers?
•	How is the final prediction obtained from multiple, overlapping prediction (I assumed by averaging, but would be good to state explicitly)
•	Images in Figure 1 appear stretched – is image anisotropy taken into account in the visualization?
•	In Fig 3, please keep the y-axis rang the same between the subfigures.


**Special Issue:**

No

---

> ### Comment · ~Shuqing_Chen1 · 2018-05-14
> **Thank you for your review**
>
> Thank you very much for your review.  Here are some replies.
>
> The comparison of DECT with SECT is described in the prior work, which is accepted by CT Meeting 2018 and the draft is preprinted by https://arxiv.org/abs/1710.05379. The prior work showed that the segmentation with optimal alpha achieved better accuracy than using one single energy image.
>
> A 9-fold cross-validation is used for the evaluation.  Section 2.3 described how the train, validation and test images were selected for one test scenario, I am sorry that it is not written so clearly.
>
> For the number of filters and the filter size, I thought it's too much for the graph... I will add them to the later version. Also, the parts of other comments will be improved later, thank you!

---

### Decision · Program_Chairs · 2018-05-15
**Paper39 Acceptance Decision**

Poster